# FREESON: RETRIEVER-FREE RETRIEVAL-AUGMENTED REASONING VIA CORPUS-TRAVERSING MCTS

## ABSTRACT

Large Reasoning Models (LRMs) have demonstrated remarkable capabilities in multi-step reasoning and calling search engines at appropriate steps. However, existing retrieval-augmented reasoning approaches rely on separate retrieval models, limiting the LRM's role in retrieval to deciding *when* to retrieve and *how* to query. This architectural separation not only imposes a dual burden of training and operational overhead for both models but also results in retrieval errors stemming from the representation bottleneck, as the retriever's limited embedding space struggles to capture the subtle distinctions required by the generator. To address this, we shift our perspective on retrieval from sequence-to-sequence matching to locating the answer-containing paths within the corpus, and propose a novel framework called **FREESON** (Retriever-**FREE** Retrieval-Augmented Rea**SON**ing). This framework enables LRMs to directly interact with external knowledge sources and autonomously acquire the information they need by acting as a unified generator-retriever. To achieve this, we introduce a variant of the MCTS algorithm specialized for the retrieval task, which we call **CT-MCTS** (**C**orpus-**T**raversing **M**onte **C**arlo **T**ree **S**earch). In this algorithm, LRMs navigate the corpus toward answer-containing regions. Experiments on five open-domain QA benchmarks covering both single-hop and multi-hop questions demonstrate that FREESON achieves an average improvement of 14.4% in EM and F1 over four multi-step reasoning models with a separate retriever, and it also performs comparably to the strongest baseline, surpassing it by 3% on PopQA and 2WikiMultihopQA, and by 12% on the fact-checking benchmark FEVER.

## 1 INTRODUCTION

Retrieval-Augmented Reasoning (RAR) is a widely used framework to reduce hallucinations and generate more factual responses by injecting external knowledge into the reasoning chain (Jiang et al., 2023; Press et al., 2023; Asai et al., 2023; Li et al., 2025a; Jin et al., 2025; Song et al., 2025; Yao et al., 2023; Wang et al., 2025; Schick et al., 2023). In such pipelines, external knowledge is crucial for guiding subsequent reasoning steps. However, conventional search engines—typically based on dual-encoder architectures—often suffer from inherent limitations, failing to retrieve appropriate documents due to an representation bottleneck, where embedding vectors cannot sufficiently represent subtle distinctions between documents or their relevance to the question (Wang et al., 2023; Kim et al., 2024; Magesh et al., 2024). For example, given the query "Where was the place of burial of John Tuchet, 6th Baron Audley's father?", $E5_{base}$ (Wang et al., 2024a) (a state-of-the-art retriever) assigns higher similarity scores to "John Tuchet, 8th Baron Audley" and "George Tuchet, 9th Baron Audley" (both incorrect) than to "John Tuchet, 6th Baron Audley" (correct), retrieving an incorrect document in the first hop *due to a single-character difference*, and consequently failing to reach the final answer document, "James Tuchet, 5th Baron Audley".

To address this issue, prior works have proposed better representation learning methods, or scaling up either the model size or the amount of training data to enhance retrieval performance (Izacard et al., 2022a; Ram et al., 2022; Wang et al., 2024a;b; Lee et al., 2024; Shao et al., 2025). However, fundamentally resolving the representation bottleneck remains challenging due to the nature of the architecture. In addition, maintaining two separate models introduces additional hardware overhead and operational costs (Zhang et al., 2024b; Reichman & Heck, 2024). In this paper, we revisit the

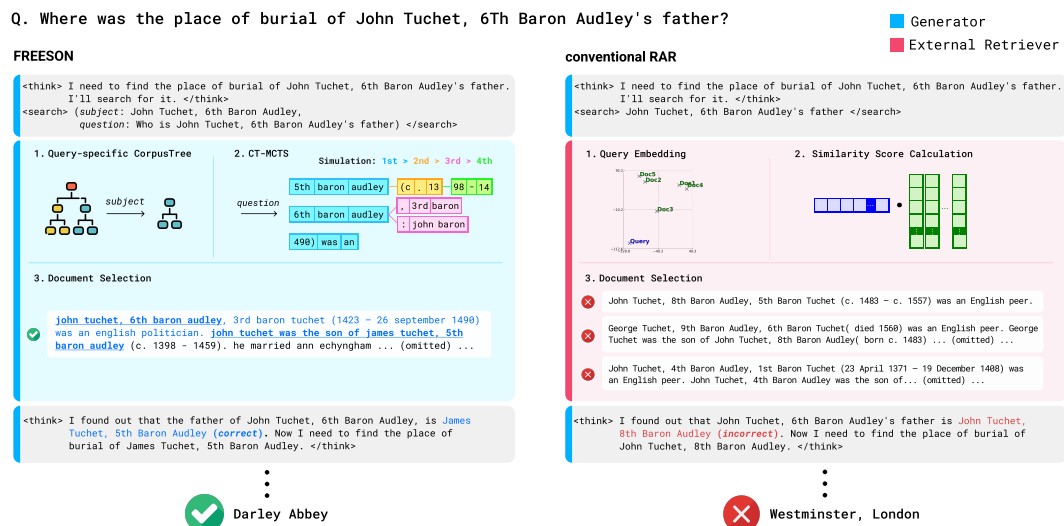

Figure 1: Overview of retrieval-augmented reasoning process. **Left:** FREESON performs both reasoning and retrieval using a single generator via CT-MCTS. **Right:** conventional RAR methods compute similarity scores between query and document embeddings using a separate retrieval model. FREESON requires neither an external retriever nor additional corpus memorization training.

conventional retrieval-augmented paradigm and pose a question: *Can a single LRM autonomously acquire the required knowledge from a corpus without relying on a separate retrieval model?*

To address this question, we shift our focus from sequence-to-sequence matching to locating the answer-containing paths within the corpus for retrieval and propose a novel framework called **FREESON** (Retriever-**FREE** Retrieval-Augmented Rea**SON**ing). In this framework, a single LRM functions as both a generator and a retriever, autonomously searching external knowledge sources without an intermediate retriever to improve the reliability of its responses. To implement this, we first introduce **CT-MCTS** (**C**orpus-**T**raversing **M**onte **C**arlo **T**ree **S**earch), a retrieval-specialized MCTS (Kocsis & Szepesvári, 2006; Silver et al., 2016; Chen et al., 2024; Liu et al., 2024; Zhang et al., 2024a), which defines its search nodes at the token level, allowing each node to represent a prefix of one or more tokens, where the prefix is constrained by a predefined index to ensure that the search follows only sequences that exist in the corpus (§ 2.2).

Implementing such retrieval-oriented MCTS introduces two key challenges: (1) the search operates at extremely fine granularity, with token-level nodes, making it difficult to capture meaningful semantics due to step-wise constraints; and (2) the model must obtain appropriate node value estimates to guide the search toward the desired location in the corpus that contains the answer. To address the first challenge, we increase node granularity while preserving single-token-level constraints, and incorporate stochastic beam search into the expansion process, enabling the LRM to actively guide expansion with multiple parallel continuations, partially taking over the role of iterative exploration. For the second challenge, we train an on-policy value network to estimate answer-containment, using CT-MCTS rollouts in environments aligned with actual inference-time scenarios (§ 4.4).

We evaluate FREESON on five open-domain QA benchmarks comprising of single-hop and multi-hop questions. On average, FREESON achieves 14.4% improvement in EM and F1 compared to four reasoning models using a separate retriever in their reasoning pipeline. It also performs on par with the strongest baseline, Search-R1, surpassing it by an average 3% on PopQA and 2WikiMultihopQA, and by 12% on the fact-checking benchmark FEVER. (§ 3.3). Our retrieval-specialized CT-MCTS plays a key role in this performance. Flexible node granularity yields a 27.5% gain over single-token nodes (§ 4.2), and multi-node expansion guided by the LRM improves performance by 13% over single-node expansion (§ 4.3). FREESON does not require any training and is applicable to an arbitrary LMs when we could access the output logit values. Particularly, it is well suited for domain-specific applications that have unlabeled corpus, as it directly explores and reasons over the content without using any external search engine.

## 2 METHODOLOGY

In this section, we detail how FREESON autonomously identifies and retrieves the knowledge it needs. See Figure 1 for an illustration of this process.

### 2.1 DYNAMICALLY ADAPTED SEARCH SPACE

When search is required, FREESON first generates a query in the form of *(subject: [subject], question: [question])*, where [subject] is the entity or proper noun the question is mainly about, and [question] is the corresponding question. Then, based on the subject information, we adaptively construct a prefix-based index we call *CorpusTree* that narrows the search space by filtering out irrelevant documents, similar to how ANN search prunes candidates in embedding retrieval.

### 2.2 CT-MCTS FOR SELF-INTERACTION WITH KNOWLEDGE

Given a question, which we input using a few-shot prompt that guides reasoning in written form (see Appendix F.2), and a filtered CorpusTree, FREESON locates the appropriate information using CT-MCTS designed to reinforce paths that are likely to contain the correct answer. Basically, CT-MCTS operates in a token-level search space where the LLM's probability distribution is dynamically masked by the CorpusTree, allowing only valid sequences found in the corpus. The CorpusTree is implemented using an FM-Index (Ferragina & Manzini, 2000; Bevilacqua et al., 2022) based on the Burrows-Wheeler Transform (BWT) (Manzini, 2001), which enables efficient and compressed prefix-constrained search. In the following, we describe the key components of CT-MCTS.

**Selection.** The first step in each simulation is to select a node from the search tree for exploration. To do so, we employ the widely used selection function, UCT (Kocsis & Szepesvári, 2006; Auer et al., 2002). Starting from the root node, we recursively select the child node $a \in \text{CorpusTree}(s)$ that maximizes the UCT score at the current node $s$, until a leaf node is reached.

$$a^* = \arg\max_a \left[ Q(s,a) + \lambda \cdot \sqrt{\frac{\log \sum_b N(s,b)}{1 + N(s,a)}} \right]$$

Here, $Q(s,a)$ is the average value of taking action $a$ from node $s$, $N(s,a)$ is the number of times action a has been selected from node $s$, and $\lambda$ is a scalar balancing exploration and exploitation. The action space is constrained by a CorpusTree according to the expansion process described below.

**Granularity-aware multi-node expansion.** The second step is to expand the selected node by determining the promising next search directions. Our expansion approach differs from conventional MCTS in two key ways.

(1) *Expanding Nodes Granularity*: Each node in our search tree contains a sequence of tokens (length $G$), rather than a single token (e.g., "5th Baron Audley" instead of "5th"; see Figure 1). This allows the model to make more context-aware and semantically meaningful decisions at each step, while also enabling faster search by traversing multiple steps in a single move. This allows the reasoning model to contribute more actively to the expansion process. Through this, we strengthen the effectiveness of CorpusTree-guided search while still adhering to token-level constraints for retrieval (§ 4.2).

(2) *Multi-node expansion per simulation*: Rather than expanding a single node per simulation, we expand the $M$ candidate children based on the LRM's next-token probabilities (e.g., ", 3rd baron" and ": john baron" are expanded in a single simulation; see Figure 1). This allows the model to better utilize the LRM's outputs during expansion, resulting in substantial performance gains (§ 4.3).

To enable the two features, we employ stochastic beam search decoding. Let $\mathcal{A}_s = \text{CorpusTree}(s)$ denote the set of valid next tokens for the current selected path $s$, and $\tilde{\mathcal{A}}_s = \{\tilde{a}_1, \ldots, \tilde{a}_k\} \subset \mathcal{A}_s$ be the top-$k$ tokens ranked by log-probability. Final candidates are sampled from $\tilde{\mathcal{A}}_s$ using multinomial sampling. For each candidate, we iteratively extend the sequence by sampling tokens until either a predefined per-node token limit $G$ is reached or no valid tokens remain ($\mathcal{A}_s = \emptyset$), while maintaining the top-$M$ paths ranked by cumulative log-probability at each step.

**Evaluating answer presence in search trajectories.** For each newly expanded node, we perform a rollout to evaluate its value using an answer presence-aware value network. The rollout follows a greedy decoding process, guided by CorpusTree at each step, and continues until either the maximum sequence length is reached or no valid tokens remain.

Once the rollout terminates, the resulting path $s_{\text{final}}$ is evaluated by the value network $\mathcal{V}$, which takes as input a prompt-style sequence $x$ consisting of the question and $s_{\text{final}}$, and outputs a value estimate. The detailed prompt format is provided in Appendix F.3. The value is computed as:

$$\hat{y} = \sigma\left(\mathcal{R}(\text{pool}(f(x)))\right)$$

where $f(x)$ denotes the decoder's final hidden states used as input to the value head, $\text{pool}(\cdot)$ extracts the hidden state of the last token, and $\sigma$ is the sigmoid function. The output scalar $\hat{y} \in [0, 1]$ serves as the value signal, which is used to update the statistics $Q(s, a)$ and $N(s, a)$ during the backpropagation phase.

### 2.3 TRAINING THE VALUE NETWORK

To evaluate whether a candidate path contains the information necessary to answer the question, we train value networks on rollouts from CT-MCTS and synthetic paths generated by LLM. In our experiments, evaluation is performed using the former. Each result is described in Section 4.4.

**On-policy training on CT-MCTS rollouts.** In this on-policy approach, we directly leverage intermediate rollouts collected during CT-MCTS execution. At each expansion step, we pause and return to the original reasoning process, feeding the current candidate path and the question into the model. The model then generates an answer, which is compared to the ground-truth to assign a soft value: 1.0 for a full match, 0.8 for a partial match, and 0.0 if there is no match. The value network is implemented by attaching a classification head to the frozen backbone of the original reasoning model and trained using binary cross-entropy loss. Training is performed on approximately 15,000 such rollout-label pairs obtained from 400 PopQA (Mallen et al., 2023) examples.

**Off-policy training on synthetic trajectories.** In this off-policy setup, we generate diverse synthetic retrieval paths and corresponding value scores using GPT-4o on the 2WikiMultihopQA dataset (Ho et al., 2020). To simulate realistic trajectories, we construct three paths per query, varying in length, relevance, and whether the final answer is entailed. Detailed prompt is in Appendix F.4. We input each query paired with its evidence sentences and generate three retrieval paths with corresponding value scores, resulting in a total of 147,755 path–value pairs. We then train a classification head on top of the frozen backbone of the original reasoning model.

### 2.4 DOCUMENT SELECTION FROM RETRIEVED PATHS

After obtaining multiple paths through CT-MCTS, we must determine how to present the identified references for downstream reasoning. We consider three possible strategies for document selection:

**Direct Path:** providing only the exact retrieved path spans.

**Window Expansion:** extending retrieved paths with surrounding context windows.

**Complete Document:** returning the complete documents from which the retrieved spans originate.

In this work, we implement the **Complete Document** approach. While direct path or window expansion may offer compact references, they risk omitting potentially important information that lies outside the selected regions or fragmenting coherent explanations. By supplying complete documents, we alleviate potential information loss. Unlike dual-encoder models that always retrieve the predefined top-$k$ documents based on similarity scores, FREESON retrieves only documents containing the search trajectories, reducing the possibility of including noisy or irrelevant information in the retrieved content (see *3. Document Selection* of Figure 1).

| Method | General QA | | | | Multi-hop QA | | | | | | Fact-checking |
|---|---|---|---|---|---|---|---|---|---|---|---|
| | PopQA | | TriviaQA | | HotpotQA | | 2WikiMultihopQA | | MuSiQue | | FEVER |
| | EM | F1 | EM | F1 | EM | F1 | EM | F1 | EM | F1 | ACC |
| *Reasoning w/o retrieval* | | | | | | | | | | | |
| Qwen2.5-7B | 0.09 | 0.11 | 0.26 | 0.31 | 0.13 | 0.19 | 0.19 | 0.23 | 0.02 | 0.06 | - |
| R1-Distill-Qwen-7B | 0.07 | 0.10 | 0.13 | 0.17 | 0.11 | 0.15 | 0.18 | 0.20 | 0.01 | 0.03 | - |
| *Retrieve-then-reasoning* | | | | | | | | | | | |
| E5 + Qwen2.5-7B | 0.13 | 0.16 | 0.15 | 0.19 | 0.10 | 0.15 | 0.19 | 0.23 | 0.02 | 0.05 | - |
| *Multi-step reasoning with external retrievers* | | | | | | | | | | | |
| FLARE | 0.20 | 0.28 | 0.29 | 0.41 | 0.21 | 0.28 | 0.27 | 0.32 | 0.06 | 0.14 | - |
| Self-Ask | 0.21 | 0.24 | 0.33 | 0.45 | 0.18 | 0.27 | 0.22 | 0.28 | 0.03 | 0.09 | - |
| Search-o1 | 0.13 | 0.15 | 0.36 | 0.43 | 0.19 | 0.25 | 0.09 | 0.12 | 0.03 | 0.10 | - |
| Search-R1 | 0.35 | 0.39 | **0.54** | **0.67** | **0.40** | **0.53** | 0.54 | 0.61 | **0.12** | **0.20** | 0.59 |
| *Multi-step reasoning via self-retrieval* | | | | | | | | | | | |
| FREESON (Ours) | **0.39** | **0.43** | 0.51 | 0.63 | 0.31 | 0.42 | **0.55** | **0.63** | 0.11 | **0.20** | **0.71** |

Table 1: Overall performance on single- and multi-hop QA, and fact-checking tasks. **Bold** indicates best, and underline indicates second-best. All models are built on 7B LMs, but FREESON uniquely operates without any external retrieval model, yet still matches or outperforms the baselines.

# 3 EXPERIMENTS

## 3.1 BENCHMARKS

We evaluate the effectiveness of FREESON on six benchmarks, including five knowledge-intensive QA datasets and one fact-checking benchmark. See Appendix 6 for detailed dataset and retrieval settings. We use EM and F1 metrics for QA, and Accuracy for fact-checking.

**General QA:** (1) POPQA (Mallen et al., 2023), a dataset constructed of factual questions centered on entities extracted from Wikipedia pages with high view counts. (2) TRIVIAQA (Joshi et al., 2017), a dataset containing complex and factoid questions collected from trivia websites and evidence passages from web documents.

**Multi-hop QA:** (1) HOTPOTQA (Yang et al., 2018), the first multi-hop QA benchmark, which consists of questions that require reasoning over multiple Wikipedia paragraphs, and includes sentence-level supporting facts. (2) 2WIKIMULTIHOPQA (Ho et al., 2020), a dataset where each question requires reasoning over two distinct Wikipedia pages corresponding to different entities, encouraging cross-page inference. (3) MUSIQUE (Trivedi et al., 2022), a dataset containing 2-4 hop questions, requiring complex reasoning (Krishna et al., 2025), curated to test compositional reasoning over multiple evidence sentences with reduced lexical overlap between questions and supporting contexts.

**Fact-checking:** (1) FEVER (Thorne et al., 2018), a fact-checking dataset for claim verification, with claims labeled as SUPPORTS, REFUTES, or NOT ENOUGH INFO.

## 3.2 BASELINES

We evaluate FREESON against strong baselines: **reasoning without retrieval**, which uses only parametric knowledge; **retrieve-then-reasoning**, which first retrieves relevant documents and then reasons; and **multi-step reasoning with external retrievers**, which performs step-by-step reasoning with interleaved retrieval. Below are the methods for multi-step reasoning with an external retriever. Details on evaluation and prompts are provided in Appendices G and F.

(1) FLARE (Jiang et al., 2023) generates reasoning steps and triggers retrieval when any token has low confidence, using a look-ahead next step as the retrieval query.

(2) SELF-ASK (Press et al., 2023) employs a scaffolded reasoning approach by generating sub-questions and corresponding intermediate answers to build the final answer.

(3) SEARCH-O1 (Li et al., 2025a) performs reasoning with interleaved retrieval and uses a separate Reason-in-Documents module when injecting retrieved documents into the reasoning chain to provide more accurate information.

(4) SEARCH-R1 (Jin et al., 2025) is trained through reinforcement learning (e.g., PPO (Schulman et al., 2017), GRPO (Shao et al., 2024b)) with retrieved-token masking to acquire the ability to interact with search engines during reasoning.

## 3.3 MAIN RESULTS

Table 1 presents FREESON' performance on six benchmarks.

**Comparison with Retrieve-then-reasoning.**  We observe that E5 + Qwen2.5-7B, which performs a single retrieval step before generation, improves performance on PopQA, where most questions can be answered with a single piece of evidence. This shows that even one-time retrieval can help in single-hop settings. However, on multi-hop QA, it does not bring meaningful gains, indicating that single-step retrieval is insufficient when multiple reasoning steps are required. In contrast, our method, FREESON, performs retrieval at each reasoning step and achieves 2 - 3× higher performance. This demonstrates the clear advantage of performing step-wise retrieval, aligned with each reasoning step when needed, for complex and multi-hop questions.

**Comparison with Multi-step reasoning with external retrievers.**  Our primary focus is on how effectively retrieval-augmented reasoning can be performed using a fully retriever-free approach. Our results show that FREESON achieves an average gain of +14.4% over four baseline models that use external retrievers during their multi-step reasoning. Specifically, it outperforms these baselines by +16.6% on PopQA, +13.5% on TriviaQA, +7.6% on HotpotQA, +28.4% on 2WikiMultihopQA, and +5.9% on MuSiQue. These results underscore that LRMs can obtain necessary knowledge without external retrieval models, by treating retrieval as a path-finding process, rather than relying on conventional embedding-based similarity search.

We have observed certain limitations in QA dataset annotations, which may lead to a slight underestimation of FREESON's performance. As discussed in Appendix E, we observed that ground-truth answers in some QA datasets often align with expressions found in documents retrieved by systems like E5, which may introduce some bias during evaluation.

## 4 ANALYSIS

### 4.1 WHY CT-MCTS OVER OTHER DECODING STRATEGIES?

We compare three decoding algorithms—greedy search, beam search, and CT-MCTS—in Table 2. The results show a consistent ranking under constrained decoding with a prefix-based index: *CT-MCTS > beam search > greedy search*.

Constrained decoding is dependent on previous decoding steps. Once the decoding path diverges in the wrong direction, it becomes impossible to recover. This makes greedy search particularly vulnerable in such environments. Beam search considers more candidates, but it remains deterministic and often suffers from early commitment, especially to the first token.

In contrast, CT-MCTS is better suited for retrieval-oriented decoding, as it enables more flexible exploration of how target information may be expressed in the corpus. Because CT-MCTS always starts from the root node and explores diverse

|  | PopQA | | 2WikiMultihopQA | |
| Decoding | EM | F1 | EM | F1 |
| --- | --- | --- | --- | --- |
| Greedy Search | 0.17 | 0.21 | 0.21 | 0.23 |
| Beam Search | 0.18 | 0.21 | 0.23 | 0.25 |
| **CT-MCTS** | **0.44** | **0.45** | **0.54** | **0.60** |

Table 2: Comparison of decoding strategies. Unlike deterministic methods, CT-MCTS achieves remarkable performance on constrained decoding.

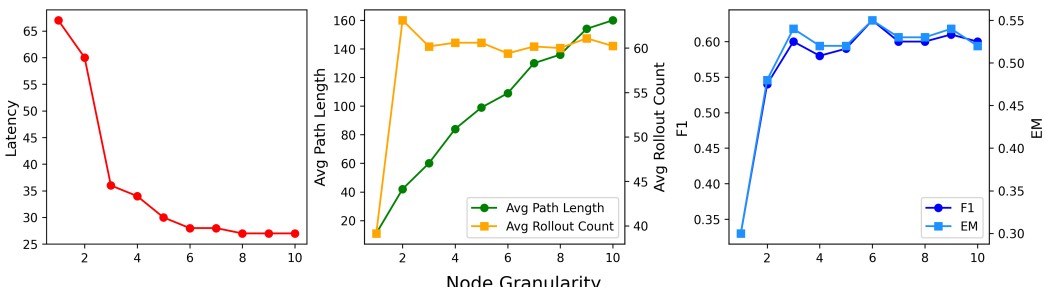

Figure 2: Impact of node granularity on system efficiency and performance (1 – 10 tokens per node). **Left**: Latency(s) sharply decreases with higher granularity, enabling faster search. **Middle**: Higher granularity yields longer, more informative paths without increasing rollout counts, indicating more efficient search. **Right**: Higher granularity leads to better performance.

|                       | Performance | | Efficiency |
|-----------------------|:-----------:|:----:|:----------:|
| **Multiple Expansion** | **EM** | **F1** | **Latency** |
| M=1                   | 0.41        | 0.46 | **21s**    |
| **M=2**               | **0.54**    | **0.60** | 31s    |
| M=3                   | 0.54        | 0.60 | 36s        |
| M=4                   | 0.54        | 0.60 | 44s        |

Table 3: Effect of the number of expanded nodes ($M$) per simulation. On 2WikiMultihopQA, performance improves significantly at $M=2$ and quickly saturates due to constrained token space.

paths from the very first token, it can more effectively follow the guidance of a value network to construct an optimal retrieval trajectory.

### 4.2 WHAT NODE GRANULARITY IS MOST EFFECTIVE FOR TRAJECTORY EXPLORATION?

To enable more context-aware decisions during CT-MCTS, we expand node granularity from a single token to multiple tokens. Our findings highlight the importance of *choosing a granularity level that preserves semantic meaning while not hindering the fine-grained adjustments of MCTS*.

As shown in Figure 2, a moderately coarse granularity ($G = 6$), corresponding to six-token nodes, achieved the best performance (F1 ↑). This level of granularity allows nodes to capture coherent semantic units, even if they do not correspond to complete linguistic phrases, while also providing significant speed improvements (Latency ↓) and longer retrieved paths (Avg Path Length ↑). These benefits arise from providing CT-MCTS with semantically richer units, which help overcome the limitations of token-level search under constrained decoding.

In contrast, at finer granularities (e.g., $G = 1$ and $G = 2$), CT-MCTS takes considerably more time, while retrieving shorter and often uninformative paths. Performance also degrades, possibly due to the increased number of simulations needed to identify optimal paths and capture semantic relationships between nodes—reflected in the notably high number of leaf nodes in Table 7, indicating scattered and uncertain search behavior.

However, overly coarse granularity is not always better. As shown by the performance decline from $G = 6$ to $G = 10$, excessively long nodes could limit fine-grained exploration, ultimately degrading performance. These results underscore *the importance of balancing semantic richness and controllability through appropriate node granularity*.

|  | PopQA | | TriviaQA | | 2WikiMultiHop | | Avg. | |
|---|---|---|---|---|---|---|---|---|
| **Value Network** | **EM** | **F1** | **EM** | **F1** | **EM** | **F1** | **EM** | **F1** |
| LLM Synthetic Trajectory | 0.41 | 0.41 | 0.48 | 0.61 | 0.53 | 0.59 | 0.47 | 0.54 |
| **CS-MCTS Rollout** | **0.44** | **0.45** | **0.51** | **0.63** | **0.54** | **0.60** | **0.50** | **0.56** |

Table 4: Comparison between CT-MCTS and LLM-based value policy. On-policy CT-MCTS performs better by learning value estimates aligned with the inference-time environment.

| | **Efficiency** | |
|---|---|---|
| **Metric** | **CT-MCTS** | **Dual-encoder** |
| Indexing time ($\downarrow$) | **63.665 mins** (cpu) | 176.855 mins (gpu) |
| Index memory ($\downarrow$) | **11GB** | 61GB |
| FLOPs ($\downarrow$) | 1.88e13 | **4.42e10** |

Table 5: Comparison of efficiency in indexing time, index memory, and FLOPs. CT-MCTS builds an FM-index, while the dual encoder uses embedding-based indexing. FLOPs are theoretical estimates without optimization; calculation details are in Appendix D.

### 4.3 CAN LLM-DRIVEN MULTI-NODE EXPANSION BOOST PERFORMANCE?

To better leverage the LLM's reasoning ability over document expressions during expansion in CT-MCTS, we explore expanding multiple candidate child nodes per simulation, selected via multinomial sampling over top-$k$ predictions from the LLM. As shown in Table 3, setting $M = 2$ achieves the best performance, improving EM and F1 scores by 13 and 14 points, respectively, compared to $M = 1$.

This improvement stems from the increased involvement of the LLM in selecting the node to expand, providing more informed guidance. However, under constrained decoding, the number of valid next-token candidates is inherently limited. As a result, although expanding more candidates is initially beneficial, the performance quickly saturates—as many of the additional nodes are likely to be explored by later simulations anyway. From $M = 3$ onward, no further gains are observed. These findings indicate that even a modest increase in LLM involvement can meaningfully improve retrieval quality in CT-MCTS.

### 4.4 CT-MCTS ON-POLICY VS. SYNTHETIC OFF-POLICY: WHICH GIVES BETTER VALUE ESTIMATES?

Assigning an appropriate value to each explored node is crucial for guiding the search toward answer-containing paths. We compare two types of value models: (1) an on-policy model trained from our CT-MCTS rollouts, and (2) an off-policy model trained on LLM-generated samples. As shown in Table 4, value models trained on CT-MCTS rollouts outperform those trained on LLM-generated data. This is likely due to stronger alignment with the actual inference-time behavior of CT-MCTS, as the value model is trained directly on the actions the system would take during real search. Additionally, this on-policy approach is cost-efficient, as it eliminates the need for separate synthetic data generation. These results suggest that training the value network within the true CT-MCTS environment is more effective, and we adopt this strategy in our method. During training, we use an 80GB A100 GPU.

### 4.5 EFFICIENCY OF CT-MCTS SELF-RETRIEVAL OVER EMBEDDING-BASED RETRIEVAL

On the 21M Wikipedia corpus, CT-MCTS indexes about $3\times$ faster (63.7 mins vs. 176.9 mins) and requires about $5.5\times$ less memory (11GB vs. 61GB), even smaller than the 14GB raw corpus due to compression with the BWT. While dual encoders store external knowledge as large embedding vectors that must be managed during inference, CT-MCTS eliminates the need for handling vectors

outside the model. However, as an inference-time scaling approach, CT-MCTS leads to higher FLOPs during inference (1.88e13 vs. 4.42e10). Unlike dual encoders whose cost scales with corpus size (Kim et al., 2024), CT-MCTS operates over valid next tokens regardless of corpus size, and its efficiency can potentially be improved with LM optimization techniques such as speculative decoding (Leviathan et al., 2023) and pruning (Sun et al., 2024).

## 5    RELATED WORKS

### 5.1    RETRIEVAL-AUGMENTED GENERATION

Large Language Models (LLMs) achieve strong performance in language understanding and generation but suffer from hallucinations in domain-specific tasks. Retrieval-Augmented Generation (RAG) alleviates this by retrieving relevant documents before generation, improving factuality (Guu et al., 2020; Lewis et al., 2021; Borgeaud et al., 2022; Izacard et al., 2022b). Early approaches follow a retrieve-then-generate framework (Lewis et al., 2021), while later work studied what (Khandelwal et al., 2020; Borgeaud et al., 2022), how (Ram et al., 2023), and when to retrieve (Guu et al., 2020).

Retrieval has since been integrated into reasoning: Self-Ask (Press et al., 2023) generates sub-questions, FLARE (Jiang et al., 2023) uses look-ahead queries, and Self-RAG (Asai et al., 2023) decides when to retrieve and evaluate content autonomously. With Large Reasoning Models (LRMs)(DeepSeek-AI et al., 2025; Zhong et al., 2024; Qwen et al., 2025), methods such as Search-o1(Li et al., 2025a) and Search-R1 (Jin et al., 2025) enhance retrieval through agentic mechanisms or reinforcement learning, while ReAct (Yao et al., 2023) directly employs external tools. FREESON extends this line by directly interacting with external knowledge, eliminating separate retrievers along with their training overhead and representation bottleneck.

### 5.2    LLMS WITH MONTE CARLO TREE SEARCH

Monte Carlo Tree Search (MCTS) (Kocsis & Szepesvári, 2006) builds a search tree through selection, expansion, rollouts, and backpropagation, and has recently been combined with LLMs to enhance reasoning. AlphaMath (Chen et al., 2024) applies MCTS for math reasoning with value models, ReST-MCTS* (Zhang et al., 2024a) leverages process reward guidance for self-training, and PPO-MCTS (Liu et al., 2024) integrates PPO value networks with MCTS at inference. To the best of our knowledge, FREESON is the first to adapt MCTS for retrieval, introducing CT-MCTS, which enables a single LRM to traverse the corpus with multi-token nodes and on-policy value estimation to locate answer-containing paths.

### 5.3    GENERATIVE INFORMATION RETRIEVAL

Generative Information Retrieval (GenIR) (Li et al., 2025b) integrates autoregressive LMs into retrieval, typically by generating document identifiers under constrained decoding with prefix-tree structures. Approaches vary by identifier format: some generate titles (Cao et al., 2021), others generate document IDs (Tay et al., 2022; Wang et al., 2023; Zeng et al., 2023; 2024), while SEAL (Bevilacqua et al., 2022) generates spans, later extended by MINDER (Li et al., 2023b) and LTRGR (Li et al., 2023a). Dynamic settings have been studied in DSI++ (Mehta et al., 2023), Corpusbrain (Chen et al., 2022), and DynamicIR (Kim et al., 2024). Recent research introduced end-to-end LLM-driven architectures unifying all IR functions within a single model by internalizing the corpus through self-supervised learning (Tang et al., 2024). FREESON extends GenIR by unifying generator and retriever at inference time without extra memorization trainings.

## 6    CONCLUSION

We revisit the conventional retrieval-augmented approach with separate retrievers and propose FREESON, where a single model serves as both generator and retriever, mitigating representation bottlenecks. To this end, we introduce CT-MCTS, a retrieval-specific search algorithm that guides LMs to traverse the corpus toward answer-containing regions. Across six tasks, our retriever-free framework shows strong performance without external search engines, demonstrating that LMs can directly leverage external knowledge to improve response reliability.

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

# APPENDIX

## A  LIMITATIONS

Our method is particularly well-suited for scenarios with a predefined corpus. In QA tasks where the corpus is not predefined, our method may be less effective compared to web-based retrieval systems, which can flexibly access a broader and more diverse range of information. Although the development of large-scale corpora, such as MassiveDS-1.4T/140B (Shao et al., 2024a), helps mitigate this limitation, efficiently handling them at scale remains a challenge.

Furthermore, our current inference-time algorithm is not explicitly optimized for reasoning over how knowledge is expressed in the corpus. Incorporating reinforcement learning techniques such as PPO to optimize the LRM's traversal over retrieval candidates could enable more adaptive and faster convergence. In addition, because CT-MCTS is an inference-time scaling algorithm, it introduces extra latency, which needs to be further optimized to make the approach more practical in real-world settings.

## B  DATASET STATISTICS & RETRIEVAL SETTINGS

Table 6 presents the statistics and retrieval settings of the five knowledge-intensive QA datasets used in our experiments.

| Settings | PopQA | TriviaQA | HotPotQA | 2WikiMultihopQA | MuSiQue | FEVER |
|---|---|---|---|---|---|---|
| | *Mallen et al. (2023)* | *Joshi et al. (2017)* | *Yang et al. (2018)* | *Ho et al. (2020)* | *Trivedi et al. (2022)* | Thorne et al. (2018) |
| *Dataset statistics* | | | | | | |
| # Examples | 500 | 500 | 500 | 500 | 500 | 495 |
| Gold answer count | Many | Single | Single | Single | Single | - |
| *Retrieval settings* | | | | | | |
| Corpus | Wikipedia-dpr | Wikipedia-dpr | Wikipedia-dpr | Wikipedia-2wiki | Wikipedia-dpr | Wikipedia-dpr |
| Corpus size | 21M | 21M | 21M | 6M | 21M | 21M |
| Retriever | E5 | E5 | E5 | E5 | E5 | E5 |
| Top-k | 2 | 3 | 3 | 3 | 3 | 3 |

Table 6: Comparison of datasets and retrieval settings. As shown in *# Examples*, to reduce computational cost, we randomly sample up to 500 examples from each dataset. In *Gold answer count*, "Many" denotes multiple gold answers; "Single" denotes exactly one.

| Node Granularity | Efficiency | | | Performance | | Exploration |
|---|---|---|---|---|---|---|
| | Latency ($\downarrow$) | Avg Rollout Count | Avg Path Length. | EM ($\uparrow$) | F1 ($\uparrow$) | Exploration |
| G=1 | 67s | 39.14 | 11 | 0.30 | 0.33 | 32 |
| G=2 | 60s | 63.13 | 42 | 0.48 | 0.54 | 16 |
| G=3 | 36s | 60.19 | 60 | 0.54 | 0.60 | 12 |
| G=4 | 34s | 60.60 | 84 | 0.52 | 0.58 | 10 |
| G=5 | 30s | 60.60 | 99 | 0.52 | 0.59 | 9 |
| G=6 | **28s** | 59.40 | 109 | **0.55** | **0.63** | 9 |
| G=7 | 28s | 60.18 | 130 | 0.53 | 0.60 | 9 |
| G=8 | 27s | 60.00 | 136 | 0.53 | 0.60 | 8 |
| G=9 | 27s | 61.10 | 154 | 0.54 | 0.61 | 9 |
| G=10 | 27s | 60.23 | 160 | 0.52 | 0.60 | 8 |

Table 7: Analysis of efficiency, performance, and exploration behavior under varying node granularity. Avg Rollout Counts for $G = 4$ and $G = 5$ are imputed with the mean (excluding $G = 1$), as single-token nodes ($G = 1$) often fail to yield valid rollouts due to limited semantic context.

## C   ANALYSIS ON VARYING NODE GRANULARITY

Table 7 shows the efficiency, performance, and exploration behavior under varying node granularity. As granularity increases, latency decreases, while the average path length grows—indicating that more informative reasoning paths are explored. Performance also improves with increasing granularity, peaking at $G = 6$ and saturating thereafter. Interestingly, although the number of valid rollouts required during the CS-MCTS process remains theoretically similar or even lower, the length of the retrieved paths increases with granularity. This is because longer current nodes provide more contextual constraints, reducing the number of valid tokens available for rollout at the next step. In other words, increased granularity allows the model to extract more informative reasoning paths with comparable or even reduced computational cost. In the case of $G = 1$, FREESON must not only search for answer-containing paths but also infer semantic relationships between adjacent nodes, leading to broader exploration across many nodes. A higher number of explorations in this setting should not be interpreted as a positive signal.

## D   FLOPS ANALYSIS

To assess computational efficiency, we measured the retrieval FLOPs of CT-MCTS in comparison to a dual-encoder retriever.

FREESON's complexity mainly depends on vocabulary size, not corpus size, and constrained decoding reduces the candidate vocabulary exponentially as decoding progresses.

In contrast, dual-encoder methods scale with corpus size as shown in Figure 4 of the paper (Kim et al., 2024). Although dual encoders benefit from FAISS indexing for lower latency, Table 3 shows that they require more FLOPs than generative retrieval methods. Our findings are consistent with this.

FREESON maintains efficiency at any corpus scale by decoding over a search space determined by the subject, using vocabulary masking throughout the process.

Following the DE FLOPs formula from Appendix A.7 of the aforementioned paper, we estimate the FLOPs of the E5 retriever using our own forward pass formulation below: (FW $\times$ number of layers) + (corpus size $\times$ (2d - 1)), with corpus size = 21M, d = 768, and number of layers = 12.

The FLOPs formula for CT-MCTS is provided below.

$$\textbf{CT-MCTS} = \sum_{i=1}^{S}\big(\text{Selection} + \text{Expansion}(V_o \cdot e^{-ki}) + \text{Rollouts}(V_o \cdot e^{-ki})\big)$$

- S represents the number of simulations. Each simulation consists of a selection, an expansion, and a rollout phase. Backpropagation is negligible in FLOPs.
- Vo is the original vocabulary size.

- exp(-k·i) represents the size of the valid vocabulary space at simulation step i, which decreases exponentially due to prefix constraints enforced by the FM-index during constrained decoding. k is the decay rate.

**Selection (per simulation)** $= D \cdot (b - 1) \cdot 7$

- 7 is decomposed as 1 for sum + 5 for UCT + 1 for max
- D represents the average depth of the selection path from the root to the selected leaf node.
- b is the average branching factor, indicating the number of available child nodes considered at each selection step.

**Expansion (per simulation)** $= \sum_{t=0}^{L} \left[ B \times \left( |V| \cdot \log |V| \cdot e^{-kt} + |V| \cdot (2d-1) \cdot e^{-kt} \right) + FW(t) \cdot n_{\text{layers}} \right]$

- FW represents the FLOPs of the forward pass per layer, and is approximated as
  $4 \cdot \text{input\_len} \cdot d^2$  (Q/K/V/O projections)
  $+4 \cdot \text{input\_len}^2 \cdot d$  (attention score computation and context aggregation)
  $+4 \cdot \text{input\_len} \cdot d \cdot d_{ff}$  (feed-forward network)
- The term exp(-k · t) models the exponential decay of the valid vocabulary size due to FM-index prefix constraints.
- |V| · log|V| corresponds to filtering candidate tokens using the FM-index, and |V| · (2d - 1) corresponds to computing the inner products for valid token logits.

**Rollouts (per simulation)** $= \sum_{t=0}^{R} \left[ |V| \cdot \log |V| \cdot e^{-kt} + |V| \cdot (2d-1) \cdot e^{-kt} + FW(t) \cdot n_{\text{layers}} \right]$

$$k = 2$$
$$V_o = 152{,}064$$
$$D = 4.8$$
$$b = 1.2$$
$$B = 2$$
$$L = 5$$
$$R = 20$$
$$S = 10$$
$$\text{avg input\_len} = 82 \text{ (2WikiMultihop)}$$
$$d = 3{,}584$$
$$d_{ff} = 18{,}944$$
$$n_{\text{layers}} = 28$$

# E  LIMITATIONS OF GROUND-TRUTH ANNOTATIONS IN QA BENCHMARKS

Although both systems generate semantically correct answers, only one may match the ground-truth due to differences in surface phrasing. For instance, E5 outputs "water" or "music", whereas FREESON (ours) produces more precise alternatives such as "fluid" or "opera"—yet only the former is recognized as correct by the ground-truth annotations.

Closer inspection reveals that the ground-truth answers often reflect the phrasing found in documents retrieved by E5. This suggests that the dataset construction process may be biased toward expressions favored by embedding-based retrieval systems, which could inadvertently advantage such models during evaluation. Below are the examples found in HotpotQA.

---

**#1. FREESON retrieves correct documents but ground-truth fails to recognize valid answers**

```
----------------------------------------------------------------
*Question: What substance is always in a trap to prevent the passage
of anything from either direction,  even sewer gases?
----------------------------------------------------------------
*Ground-truth answer: water
----------------------------------------------------------------

<think>I need to find the substance that is always in a trap to
prevent the passage of anything from either direction, even sewer
gases. I'll search for it.</think>

<search> (subject : trap, question : What substance is always in a
trap to prevent the passage of anything from either direction, even
sewer gases) </search>

<information>For trap : Doc 0. trap (plumbing) trap (plumbing) in
plumbing, a trap is a device shaped with a bending pipe path to retain
 fluid to prevent sewer gases from entering buildings while allowing
waste to pass through. in oil refineries, traps are used to prevent
hydrocarbons and other dangerous gases from escaping through drains.
... (omitted) ...

Doc 1. trap (plumbing) venting. certain drum-styled traps are also
discouraged or banned. trap (plumbing) in plumbing, a trap is a device
 shaped with a bending pipe path to retain fluid to prevent sewer
gases from entering buildings while allowing waste to pass through.
... (omitted) ...  </information>

<think>I found out that in plumbing, a trap is a device shaped with a
bending pipe path to retain fluid to prevent sewer gases from entering
 buildings while allowing waste to pass through. Now I need to find
the substance that is always in a trap to prevent the passage of
anything from either direction.</think>

... (omitted) ...

----------------------------------------------------------------
<answer> fluid </answer>
----------------------------------------------------------------
```

#2. FREESON retrieves correct documents but ground-truth fails to recognize valid answers

```
----------------------------------------------------------------
*Question: What type of art does The Consul and Arlecchino have in
common?
----------------------------------------------------------------
*Ground-truth answer: music
----------------------------------------------------------------

#First step
<think>I need to find the type of art that The Consul and Arlecchino
have in common. I'll search for it.</think>

<search> (subject: The Consul, question: What type of art does The
Consul have) </search>

<information>For The Consul : Doc 0. the consul the consul the consul
is an opera in three acts with music and libretto by gian carlo
menotti, his first full-length opera. ... (omitted) ...

Doc 1. the consul prize for music and also the 1950 new york drama
critics' circle award for best musical. ... (omitted) ...

Doc 2. the consul the broadway cast, this time with rosemary kuhlmann
as the secretary of the consulate. ... (omitted) ... noted the
topicality of the opera by analogy to the real-life situations </
information>

#Second step
<think>I found out that The Consul is an opera. Now I need to find the
 type of art that Arlecchino has in common with The Consul.</think>

<search> (subject: Arlecchino, question: What type of art does
Arlecchino have) </search>

<information>For Arlecchino : Doc 0. arlecchino (opera) of a double-
bill. the first british staged performance of arlecchino was in 1954
at glyndebourne. however, edward clark had produced a concert version
in london in 1939. the opera is in four movements ... (omitted) ...

Doc 1. arlecchino (opera) arlecchino, oder die fenster (harlequin, or
the windows, is a one-act opera with spoken dialog by ferruccio busoni
. ... (omitted) ...

Doc 2. arlecchino (opera) (finale.) in procession the other characters
, leandro and colombina, dottore and abbate, the donkey and the cart,
the two constables, and finally arlecchino and annunziata, cross the
stage and bow to the audience ... (omitted) ... </information>

<think>I found out that Arlecchino is an opera. Now I can compare the
type of art that The Consul and Arlecchino have in common. Both The
Consul and Arlecchino are operas.</think>

----------------------------------------------------------------
<answer> opera </answer>
----------------------------------------------------------------
```

# F   PROMPT TEMPLATE FOR FREESON

## F.1   REASONING PROMPT TEMPLATE

---
**Reasoning process**

```
Answer the given question.
You must conduct reasoning inside <think> and </think> every time you
get new information.
After reasoning, if you find you lack some knowledge, you can call a
search engine by:
<search> (subject : Help! Help! Police!, question : Who is the
director of the film Help! Help! Police!) </search>
This is the correct form for the query: 'Who is the director of the
film Help! Help! Police?'
It will return the searched results between <information> and </
information>.
You can search as many times as you want.
If you find no further external knowledge is needed, you can directly
provide the answer inside <answer> and </answer>,
without detailed illustrations. For example: <answer> Beijing </answer
>
Only respond to the final question. Your answer must reflect the end
goal, not just a part of the process.
Question: {question}
```
---

## F.2   RETRIEVAL PROMPT TEMPLATE

---
**Retrieval prompts for reasoning over document structure**

```
Given a subject and a question, generate a word or phrase likely to
appear in a document
that answers the question.

Q: subject: Star Wars, question: who did Star Wars direct?
A: Star Wars is directed by

Q: subject: Alice, question: When was Alice born?
A: Alice (January 1, 1970 ~ December 12, 2024)

Now your tern:
Q: {question}
A:
```
---

## F.3 CT-MCTS VALUE NETWORK PROMPT TEMPLATE FOR TRAINING AND INFERENCE

---

**Training on-policy value networks and evaluation using them**

```
# training
Answer the given question.
you can call a search engine using <search> and </search>.
It will return the top searched results between <information> and </
information>.
Based on the provided information, provide the final answer inside <
answer> and </answer>.
Question: {question}

# inference
Score from 0 to 1 how much the generated reference contains at least a
 partial answer to the query.
Query: {query_text}
Generated reference: {rollout_text}
Score:
```

---

## F.4 PROMPT TEMPLATE FOR TRAINING VALUE NETWORKS WITH SYNTHETIC ROLLOUTS

---

**Training off-policy value networks**

```
You are helping build a dataset for a reward model.\n\n
Given:\n
- A user query\n
- A reference sentence that correctly answers it\n\n
Your task:\n
1. Generate 3 diverse outputs that vary in:\n
    - Whether they contain the exact answer\n
    - Helpfulness in answering the query\n
    - Length and form (sentence, phrase, or word)\n
2. Include at least 1 short or fragment-style response.\n\n
Each output should be a dictionary with:\n
- 'generated': the output\n
- 'has_answer_score': 1 only if it contains the exact answer
textually (not paraphrased)\n
- 'sim_seq_score': float (0.0-1.0) based on how well it answers
the query\n\n
Example:\n
Query: What nationality is Aleksandr Stolper?\n
Reference: Aleksandr Borisovich Stolper (12 August 1907, Dvinsk (
now Daugavpils) – 12 January 1979, Moscow) was a Russian/Soviet
film director and screenwriter.\n
Output: [{{\"generated\": \"Aleksandr Borisovich Stolper\", \"
has_answer_score\": 0, \"sim_seq_score\": 0.7}},
{{\"generated\": \"Aleksandr Borisovich Stolper (12 August 1907,
Dvinsk (now Daugavpils))\", \"has_answer_score\": 0, \"
sim_seq_score\": 0.85}},
{{\"generated\": \"Russian/Soviet film director and screenwriter
.\", \"has_answer_score\": 1, \"sim_seq_score\": 0.7}}]\n\n
Now do the same for:\n
Query:\n{query}\n\nReference:\n{reference}\n\nOutput:
```

---

## G   EVALUATION DETAILS

All baseline LMs use Qwen2.5-7B. The FREESON framework performs retriever-free inference based on the SearchR1-nq_hotpotqa_train-qwen2.5-7b-em-ppo checkpoint from Search-R1 (Jin et al., 2025).

Hyperparameters for all models are set as follows:

- Temperature: 0.7
- Max new tokens per reasoning step: 1,024

For <retrieval> steps in FREESON:

- Max simulation: 20
- Max rollout length: 30

## H   USE OF LARGE LANGUAGE MODELS

We state that LLMs were employed in the writing process of this paper for polishing.

